# Unveiling Cryptic *BCR-ABL1* Rearrangements: Diagnostic Challenges and Clinical Impact in Myeloid Malignancies

**DOI:** 10.3390/ijms26188812

**Published:** 2025-09-10

**Authors:** Anna Ferrari, Chiara Salvesi, Eugenio Fonzi, Barbara Giannini, Michela Tonelli, Irene Zacheo, Matteo Paganelli, Federico Lo Schiavo, Marco Rosetti, Giorgia Simonetti, Giovanni Marconi

**Affiliations:** 1Biosciences Laboratory, IRCCS Istituto Romagnolo per lo Studio dei Tumori (IRST) “Dino Amadori”, 47014 Meldola, FC, Italy; anna.ferrari@irst.emr.it (A.F.); matteo.paganelli@irst.emr.it (M.P.); federico.loschiavo@irst.emr.it (F.L.S.); giorgia.simonetti@irst.emr.it (G.S.); 2Unit of Biostatistics and Clinical Trials, IRCCS Istituto Romagnolo per lo Studio dei Tumori (IRST) “Dino Amadori”, 47014 Meldola, FC, Italy; eugenio.fonzi@irst.emr.it; 3U.O. Genetica Medica, Laboratorio Unico-AUSL della Romagna, 47522 Pievesestina di Cesena, FC, Italy; barbara.giannini@auslromagna.it (B.G.);; 4Hematology Unit, IRCCS Istituto Romagnolo per lo Studio dei Tumori (IRST) “Dino Amadori”, 47014 Meldola, FC, Italy; irene.zacheo@irst.emr.it; 5Unit of Clinical Pathology, Hub Laboratory, AUSL della Romagna, 47522 Pievesestina di Cesena, FC, Italy; marco.rosetti@auslromagna.it; 6Hematology Unit, University of Bologna, S. Maria delle Croci Hospital, 48121 Ravenna, RA, Italy; giovanni.marconi@unibo.it

**Keywords:** *BCR-ABL1*, fusions, acute myeloid leukemia, diagnostics, NGS

## Abstract

Chromosomal *BCR-ABL1* fusions are the defining molecular lesions of chronic myeloid leukemia (CML) and Philadelphia-positive acute lymphoblastic leukemia, and are rarely observed in acute myeloid leukemia. Their detection have transformed treatment paradigms by enabling effective use of specific tyrosine kinase inhibitors (TKIs). Although many *BCR-ABL1* rearrangements are identified by standard cytogenetics, a clinically relevant subset is cryptic and can escape detection. High-depth RNA sequencing assays have improved our capacity to detect expressed fusion transcripts. Here, we introduce two myeloid cases in which cryptic *BCR-ABL1* rearrangements and precise breakpoints detection required an integrated molecular approach: we describe the initial diagnostic pitfalls, detail the downstream therapeutic and prognostic implications and offer practical recommendations for integrating targeted sequencing and cytogenetics into routine practice. In the first case, a patient initially diagnosed with a myelodysplastic/myeloproliferative neoplasm was reclassified as CML following the discovery of a cryptic e13a2 *BCR*-*ABL1* rearrangement, enabling effective TKI treatment. In the second case, a previously undetected *BCR*-*ABL1* fusion was identified in a relapsed AML patient, along with additional molecular lesions, underscoring the aggressive nature of the disease. Our findings support a systematic, multimodal screening strategy in patients with atypical presentations to ensure the timely detection of clinically actionable fusion events.

## 1. Introduction

The integration of molecular and cytogenetic data into routine practice has markedly improved patient diagnosis and prognostic assessment, resulting in improved treatment outcomes. Accurate diagnosis is paramount in both chronic myeloid leukemia (CML) and acute myeloid leukemia (AML), as it directly influences prognosis and guides therapeutic strategies. In CML, the identification of the *BCR-ABL1* fusion gene, resulting from the t(9;22)(q34;q11) translocation known as the Philadelphia chromosome (Ph) [1,2], has been a groundbreaking discovery that led to the development of targeted therapies like tyrosine kinase inhibitors (TKIs), drastically improving patient outcomes [3]. The Ph translocation leads to the constitutive activation of the ABL1 tyrosine kinase, which plays a central role in the pathogenesis of several hematologic malignancies, including CML, B-cell acute lymphoblastic leukemia (B-ALL), mixed-phenotype acute leukemia and, in rare cases, AML [4,5,6,7]. In AML, the identification of the translocation is useful for prognosis and treatment [8]. *BCR-ABL1* generates different fusion protein isoforms depending on the breakpoint in the *BCR* gene [breakpoint in *ABL1* is between exons (ex) 1b and 2]. The most common isoforms include p190 (e1a2), p210 (e13a2 or e14a2), and p230 (e19a2). The p210 isoform is predominant in CML and is detected in approximately 20–30% of *BCR-ABL1*-positive B-ALL cases. The p190 isoform is more frequently associated with B-ALL, while the p230 isoform is typically linked to neutrophilic CML [9,10,11]. In CML 2–5% of patients present atypical Ph+ isoforms, that are also found in about 7.5% of adult ALL (e8a2/1, e13a3/2, e14a3, e1a2/3, e6a2, e19a2, and e12a2) [12,13,14]. Whilst routine diagnostics revolutionized patients’ treatment in hematology, it still falls short in identifying large genomic events [15], cryptic events [16,17] and non-genomic drivers [18].

We here report the results obtained by integrating cutting edge methodologies with diagnostic routine, that enabled the identification of two cryptic *BCR-ABL1* fusions that were not recognized by standard chromosome banding analysis (CBA). Our effort supports the need of continuous advancement to improve the identification of diagnostic, prognostic, and druggable events that play a pivotal in patients’ management.

## 2. Case 1: The Identification of Cryptic *BCR-ABL1* Modified the Diagnostic Classification

A 61-year-old woman with a longstanding history of severe arterial hypertension presented for hematological evaluation due to persistent fatigue. She was initially diagnosed at another institution, with a myelodysplastic/myeloproliferative neoplasm (MDS/MPN) with features suggestive of early-stage myelofibrosis. She had received hydroxyurea for disease management.

At diagnosis, laboratory tests revealed leukocytosis (white blood cells [WBC] 23,000/mm^3^; neutrophils 17,620/mm^3^), 12.3 g/dL hemoglobin and 345,000/mm^3^ platelets. Bone marrow biopsy showed 5% CD34^+^ blasts, dysplastic micro-megakaryocytes, and mild reticulin fibrosis (MF-1). Cytogenetic analysis demonstrated a normal female karyotype, and at molecular testing, the patient was negative for *JAK2* and *FIP1L1-PDGFRA* rearrangements.

She was thereafter re-evaluated at our institution. Bone marrow cytogenetics via CBA revealed a del(5)(q13q22) in 4 out of 20 metaphases (46,XX,del(5)(q13q22)[4]/46,XX [16]; Figure 1A), which is not a commonly deleted region in MDS 5q- [16,19,20]. Moreover, she had an *ASXL1* frameshift mutation with a variant allele frequency (VAF) of 3.5% (Myeloid Solution panel, 31 genes, Sophia DDM software, Sophia Genetics, Rolle, Switzerland). By December 2023, 8 months later, the deletion was persisting in 5 out of 20 metaphases, and the *ASXL1* mutation VAF had increased to 9.3%.

RNA sequencing using the TruSight RNA Pan-Cancer Panel (1385 genes; Illumina, San Diego, CA, USA), routinely performed for research purposes, detected a *BCR*-*ABL1* e13a2 fusion transcript (*BCR*, NM_004327.3, chr 22: 23631808, ex13; *ABL1*, NM_005157.4, chr 9: 133729451, ex2; grch37). Using our custom-built 4-fusion-tool-pipeline specifically optimized for detecting gene fusions in acute leukemias [21], the presence of the *BCR-ABL1* fusion transcript was consistently identified by all four analytical tools and further confirmed by qualitative and quantitative RT-PCR [EasyPGX ready BCR-ABL Fusion (Diatech, Chiyoda-ku, Tokyo); BCR-ABL P210 ELITe MGB Kit (Elitech, San Jose, CA, USA)]. Copy number variation analysis (Extended Myeloid Solution, 98 full coding genes; Sophia DDM 7.0.0 software; Sophia Genetics) identified a heterozygous deletion involving exon and intron 1 of *ABL1* [NM_007313, *ABL1*_ex1—*ABL1*_int1, chr9:(133589655–133710966), 0.12 Mb–47 Mb); Appendix A]. Notably, FISH analysis revealed a *BCR-ABL1* fusion signal exclusively on chromosome 9, consistent with a cryptic insertion (Figure 1B); notably, the FISH was positive on euploid cells and on cells with del 5q, suggesting the founding role of *BCR-ABL1* in patients’ disease. The patient was thereafter correctly diagnosed as a CML, treated with tyrosine kinase inhibitor achieving molecular response (MR) 4 after 6 months of treatment.

This case underscores the diagnostic challenges posed by cryptic *BCR-ABL1* rearrangements in CML, highlighting the importance of including upfront FISH analysis and using supplemental advanced molecular techniques for accurate diagnosis and appropriate therapeutic intervention.

## 3. Case 2: Therapeutic Implications of Cryptic *BCR-ABL1* Identification

A 53-year-old woman was admitted to our institution with a history of AML initially diagnosed 2 years before, characterized by a *MECOM* rearrangement. Her clinical course included initial treatment with liposomal daunorubicin/cytarabine induction therapy, achieving partial remission with 7% blasts. After re-induction therapy, she achieved further blasts reduction to 5% and subsequently underwent a haploidentical hematopoietic cell transplant (HCT) from her son following Busulfan-tiothepa-fludarabine myeloablative conditioning regimen without significant graft-versus-host disease.

Unfortunately, the patient experienced disease relapse within 20 months after HCT. Chimerism analysis at relapse demonstrated 97% recipient origin, confirming that the clonal expansion derived from the original disease. Salvage therapies included four cycles of azacitidine plus venetoclax followed by donor lymphocyte infusions, and a trial of adoptive immunotherapy (aploCIK), both resulting in progressive disease. She was subsequently enrolled in a clinical trial utilizing menin inhibition, with progressive disease after three cycles.

When firstly evaluated at our center, laboratory findings showed leukocytosis (WBC 21,250/mm^3^), anemia (Hb 9.8 g/dL), and normal platelet count (303,000/mm^3^). Peripheral blood smears revealed significant blast expansion (66%). Bone marrow biopsy confirmed hypercellularity with 70–75% replacement of the marrow elements by blast cells, that stained positive for CD34/CD33/CD13/CD117/HLA-DR at immunophenotype analysis.

Standard cytogenetic analysis at our center revealed a normal karyotype (46,XX [20]). RNA sequencing analysis (TruSight RNA Pan-Cancer Panel, Illumina) with a 4-tool call revealed no evidence of *MECOM* rearrangement at transcript level at this time point. However, it uncovered a cryptic *BCR-ABL1* rearrangement, confirmed as the p210 isoform (e13a2) by both qualitative and quantitative RT-PCR (Figure 2A), that had not been tested at the pre-transplant stage. An extra breakpoint was detected, with one tool, between *ABL1* and *BCR* (*ABL1*, NM_005157.4, chr 9: 133589842, ex1; *BCR*, NM_004327.3, chr 22: 23632525, ex14; grch37, Figure 2A). Additional mutations, including *SF3B1* (p.His662Gln; VAF 43.6%), *NRAS* (p.Gly13Asp; VAF 46.7%) and *WT1* (p.Ala382Glyfs*3; VAF 45.5%), and several copy number alterations defining a potential 7q deletion (Extended Myeloid Solution, Sophia Genetics; Appendix A) confirmed the aggressive nature of the disease. Given her complex treatment history and molecular profile, the patient was in palliative care when we discovered the *BCR-ABL1* fusion; thus, we do not have clinical data on TKIs effectiveness in this case.

This case highlights the significance of baseline FISH analysis, and if indicated, comprehensive molecular testing, including RNA sequencing, in AML cases with atypical presentations, enabling the identification of cryptic rearrangements that are pivotal for prognostication and therapeutic decision-making.

## 4. Discussion

The results presented in this study underline the critical importance of comprehensive molecular diagnostics in hematological malignancies, particularly regarding cryptic genetic alterations such as the *BCR-ABL1* rearrangement. In the routine clinical practice, standard cytogenetics may fail to identify cryptic genetic changes, underscoring the need for FISH analysis supplemented by more advanced and high-depth molecular techniques, as RNA sequencing, likely complemented by data validation with conventional approaches (e.g., FISH, RT-PCR) to achieve an accurate disease definition and classification.

The case of the patient initially misdiagnosed as MDS/MPN with suspicion of early-stage myelofibrosis highlights the diagnostic challenges encountered when relying solely on traditional chromosome banding analysis and standard DNA-based NGS approaches. Conversely, a comprehensive RNA-based molecular profiling revealed a cryptic e13a2 *BCR-ABL1* rearrangement, changing the patient diagnosis into CML. Accurate diagnosis in such a scenario is crucial, as targeted TKIs can substantially improve the outcomes of patients with *BCR-ABL1*-positive diseases.

Similarly, the second case, initially categorized as high-risk AML with a t(3;3) translocation, emphasizes the clinical implications of identifying druggable molecular targets. Despite intensive chemotherapy and a haploidentical transplant, the patient relapsed, showing resistance to multiple lines of therapy, including novel immunotherapies. An extended molecular profiling via RNA-seq was able to uncover a cryptic *BCR-ABL1* rearrangement. While *BCR-ABL1* is a hallmark of CML and B-ALL, it is also rarely present in AML, with a reported incidence of 0.5–3% in newly diagnosed cases [22].

AML with *BCR-ABL1* is now recognized as a distinct, rare AML entity that must be differentiated from CML in myeloid blast phase; both World Health Organization 2022 and the International Consensus Classification 2022 retain a ≥20% blast requirement specifically for this genotype to minimize misclassification, and diagnostic work-up should exclude prior or concurrent CML features (e.g., basophilia, splenomegaly, cytogenetic “CML-type” additional abnormalities). Incidence is very low—typically <1% of de novo AML—and historical series consistently describe aggressive behavior with poor responses to conventional AML chemotherapy [23].

For AML, precise classification based on cytogenetic and molecular abnormalities is essential, as it determines risk stratification and informs the choice of treatment modalities, which can range from intensive chemotherapy to targeted agent combinations. Thanks to the increasing use of NGS technologies, it is now possible to identify increasingly specific and rare subgroups of patients affected by hematological malignancies, thus improving diagnosis and suggesting novel potential treatment targets, with a potential clinical value in particular in the relapsed/refractory population [24,25,26]. While TKIs such as imatinib, dasatinib, and ponatinib have demonstrated efficacy in CML and Ph+ ALL, their use in AML with *BCR-ABL1* is not well established mainly due to rarity [27,28,29]. While TKIs given as single agents TKI produced only short-lived hematologic response in *BCR-ABL1* AML [27], interestingly, in a 2024 multicenter study of 5819 AML cases, de novo *BCR-ABL1*-positive AML treated with imatinib and intensive chemotherapy reported markedly improved outcomes, including durable remissions without transplant in some patients, challenging routine assignment to adverse-risk categories [29].

A growing, albeit still limited, body of evidence supports combination strategies that layer a TKI onto venetoclax (plus HMA) for *BCR-ABL1*-positive myeloid disease.

Higher remission rates were achieved in a heterogeneous cohort of Ph+ AML and CML-blast phase (BP) patients when TKIs are combined with venetoclax-based backbones versus TKI alone. Venetoclax plus TKI (often ponatinib) produced 43% and 75% of responses in Ph+ AML and CML-BP, respectively. Mechanistically, TKIs can downregulate anti-apoptotic buffers such as MCL-1/BCL-XL and increase BCL-2 dependency, providing a rationale for synergy with venetoclax [30,31,32]. Few evidence are also available from individual Ph+ AML cases, reporting durable remissions in response to azacitidine/venetoclax plus dasatinib or to venetoclax combined with newer TKIs such as flumatinib [33,34,35]. Moreover. markedly improved outcomes were achieved in young/fit patients by intensive chemotherapy plus a TKI (e.g., imatinib) compared with chemotherapy alone [29]. However, due to the rarity of this entity, standardized treatment protocols are still missing, highlighting the need for further clinical research. Of note, *MECOM* rearrangement is a known mechanism of evolution of CML, related to a high risk of blast phase [36]. Given its aggressive nature and suboptimal response to standard AML therapies, AML with *BCR-ABL1* represents a high-risk entity requiring targeted therapeutic strategies [29,37]. The integration of molecular profiling and personalized treatment approaches is crucial to improve the outcomes of this rare but clinically significant leukemia subtype.

In this context, FISH analysis represents a crucial diagnostic tool for identifying diagnostic-based rearrangements. Unlike conventional karyotyping, which may fail to reveal subtle or hidden translocations, FISH offers a sensitive and widely available approach that can rapidly detect recurrent abnormalities such as *BCR-ABL1,* though with the limitation that in some, not infrequent, cases, the test may result not evaluable. Its inclusion in the upfront diagnostic work-up, especially in cases with unusual features or inconclusive cytogenetics, may prevent diagnostic delay and guide timely therapeutic intervention. However, it should be stressed that, compared with comprehensive genome screening, FISH cannot be fully automated and consume a significant working time of technicians and biologists. Thus, application of FISH is usually reserved to few, common and very significant fusions and deletions and cannot easily be extended to very rare ones.

Our results clearly demonstrate that druggable molecular alterations can be hidden from conventional diagnostic methods, potentially delaying or preventing the administration of the most appropriate therapy. Cryptic rearrangements like *BCR-ABL1* can profoundly impact treatment strategies, especially considering the availability and proven efficacy of TKIs. Thus, the implementation of comprehensive genomic and transcriptomic testing should be promoted, particularly in cases of atypical disease presentation or treatment-resistant leukemia. Moreover, our data reveal that hematological malignancies may harbor multiple subtype-defining genetic abnormalities, such as *MECOM* rearrangements or del(7q) in association with *BCR-ABL1*, which can occur concurrently or sequentially during the disease course. Therefore, the identification of one genetic alteration, especially if not druggable, should not be considered the arrival point of the diagnostic workup, but rather prompt further comprehensive molecular analyses to fully characterize the disease and uncover additional, potentially targetable aberrations. Hematological malignancies are characterized by a high degree of clonal heterogeneity, where distinct subclones can influence disease progression, treatment response, and relapse risk. In AML, a better understanding of clonal dynamics helps prognostication, identification of resistance mechanisms, and improvement of minimal residual disease metrics [38,39].

In conclusion, these cases reinforce the growing need for integrating advanced diagnostic methodologies in hemato-oncology to unveil hidden genetic alterations. The identification of druggable molecular targets has not only diagnostic but, more importantly, significant therapeutic relevance, emphasizing the shift toward personalized medicine and tailored treatment approaches in hematological malignancies.

## Figures and Tables

**Figure 1 ijms-26-08812-f001:**
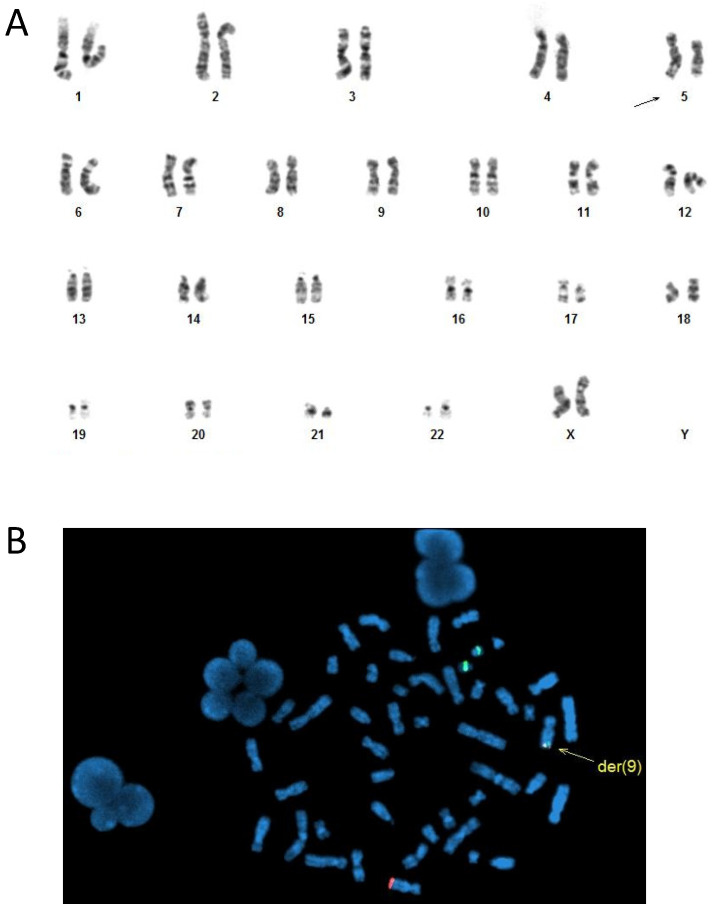
**Bone marrow karyotype and fluorescence in situ hybridization assay of case 1.** (**A**) *Representative G-banded karyotype***.** Metaphase from bone marrow cells of a female patient representing an interstitial deletion (single arrow) of the long arm of chromosome 5, del(5)(q13q22). This deletion was observed in 4 of 20 metaphases. (**B**) *FISH analysis using LSI BCR/ABL Dual Color Dual Fusion Translocation Probe (Vysis)*. Metaphase, from a leukemic sample, hybridized with ABL1 (red) and BCR (green) probes. One fusion signal is observed on a derivative chromosome 9 [der(9)], indicated by the yellow arrow, consistent with a cryptic translocation t(9;22). The presence of a single fusion signal suggests the loss of the red signal ABL1 on der(22) supporting an atypical signal pattern.

**Figure 2 ijms-26-08812-f002:**
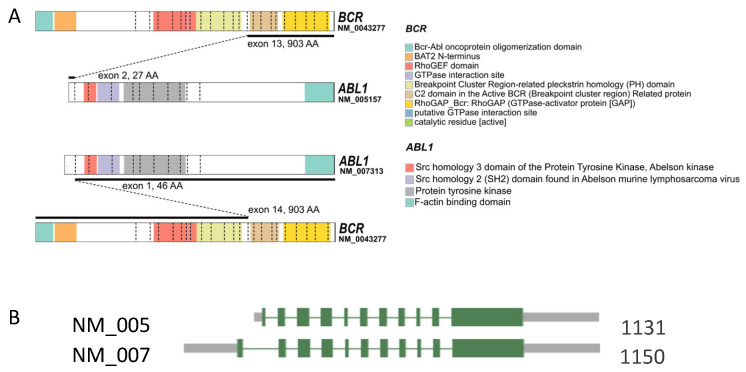
***BCR-ABL1* and *ABL1-BCR* identified fusion diagrams in case 2.** (**A**) Schematic representation of two fusion protein breakpoints identified by RNA sequencing approach. The upper fusion represents domain annotations and in frame p210 fusion scheme between *BCR* exon 13 and *ABL1* exon 2; the panel below represents domain annotations of the reciprocal fusion *ABL1-BCR*. (**B**) Representation of the two *ABL1* isoforms identified in this patient. The NM_007313 *ABL1* isoform shows an alternative exon 1 compared to the canonical one (NM_005157). Proteinpaint tool (Release version: 2.129.6-780b5acb0; Human hg19).

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
