# Peer review of "Unveiling Cryptic BCR-ABL1 Rearrangements: Diagnostic Challenges and Clinical Impact in Myeloid Malignancies"

_ijms, 2025, doi:10.3390/ijms26188812_

Round 1

Reviewer 1 Report

Comments and Suggestions for Authors

The manuscript presents two illustrative cases demonstrating how advanced molecular techniques can uncover cryptic BCR-ABL1 rearrangements missed by standard diagnostics and directly impact clinical management. The manuscript is well written and methodologically robust. I have no revision to suggest.

Reviewer 2 Report

Comments and Suggestions for Authors

This is an interesting case report for two important cases. Cryptic rearrangements for receptor tyrosine kinase genes have been under investigated in many cancer cases. These two cases stressed the importance of BCR::ABL1 FISH and RNA sequencing in work-ups for hematological malignancies. 

For case 2, was the BCR::ABL1 tested on the pre-transplant specimens? Also, we assume the relapse was from the original disease from the recipient, but it would be helpful to provide chimerism to confirm. 

For case 1, was the cryptic BCR::ABL1 rearrangement related to the 5q deletion? Also, the authors should point out that the del(5)(q13q22) is not a commonly deleted region in MDS 5q- (PMID 36428627, 22370328). 

For both cases, it would be helpful to provide marrow morphology pictures. 

For case 1, the megakaryocytes were described as "dysplastic" (Page 2 Line 58). However, megakaryocytes are usually described as dwarf megakaryocytes (or micromegakaryocytes). 

For case 2, it would be helpful to see the pre-transplant and post transplant relapse marrow image. 

For Case 2, per the NGS panels presented in Table S2, the NRAS G13D and WT1 frameshift mutations were significant findings and should be mentioned in the main text, together with the SF3B1. 

For both cases, please provide the reference genome assembly for the genomic coordinators (Page 2 Line 70 and Page 4 Lines 118/119). 

Reviewer 3 Report

Comments and Suggestions for Authors

The authors present two interesting cases of delayed diagnosis of CML due to cryptic rearrangements. In one case clarification of the diagnosis and initiation of TKI therapy resulted in significant clinical benefit to the patient. The manuscript is well-written and the data are clearly presented.

Line 93: The authors state “This case underscores the diagnostic challenges posed by cryptic BCR-ABL1 rearrangements in CML, highlighting the necessity of advanced molecular techniques for accurate diagnosis and appropriate therapeutic intervention”.

The rearrangement was detected with routine FISH analysis; therefore I would not state that advanced molecular tests were necessary. Instead I would change the statement to “This case underscores the diagnostic challenges posed by cryptic BCR-ABL1 rearrangements in CML, highlighting the importance of including upfront FISH analysis and using supplemental advanced molecular techniques to clarify the diagnosis”.

Line 96: was FISH analysis performed at any point during Case 2?

Line 112: what was the immunophenotype of the blasts in Case 2?

Line 124: The authors state “This case highlights the significance of comprehensive molecular testing, including RNA sequencing, in AML cases with atypical presentations, enabling the identification of cryptic rearrangements that are pivotal for prognostication and therapeutic decision-making”.

I would change this to “This case highlights the significance of upfront FISH analysis, and if indicated, comprehensive molecular testing, including RNA sequencing, in AML cases with atypical presentations, enabling the identification of cryptic rearrangements that are pivotal for prognostication and therapeutic decision-making”.

Line 140

The authors state” “In the routine clinical practice, standard cytogenetics may fail to identify cryptic genetic changes, underscoring the need of more advanced and high depth molecular techniques, as RNA sequencing, likely complemented by data validation with conventional approaches (e.g. FISH, RT-PCR) to achieve an accurate disease definition and classification.” This should be changed to: “In the routine clinical practice, standard cytogenetics may fail to identify cryptic genetic changes, underscoring the need for upfront FISH analysis supplemented by more advanced and high depth molecular techniques, such as RNA sequencing, to achieve an accurate disease definition and classification”.

Line 155: The authors state “An extended molecular profiling via RNA-seq was solely able to uncover a cryptic BCR-ABL1 156 rearrangement”. It appears that this patient did not undergo FISH testing which might have detected the rearrangement. Therefore I would change the statement to: “An extended molecular profiling via RNA-seq was able to uncover a cryptic BCR-ABL1 rearrangement.

The authors should add to the discussion section a few sentences describing the benefit of FISH analysis in cases of cryptic rearrangements.

Reviewer 4 Report

Comments and Suggestions for Authors

This paper is very interesting, as it discusses the diagnostic challenges and clinical impact of rare BCR-ABL1 rearrangements in CML, ALL, and AML. However, the following points would be helpful: (1) Comprehensive molecular diagnosis is crucial for hematopoietic malignancies, especially for occult genetic alterations such as BCR-ABL1 gene rearrangements. How frequent are AML, CML, and ALL cases, respectively, that warrant more advanced, in-depth molecular diagnostic techniques such as RNA sequencing? (2) While TKIs such as imatinib, dasatinib, and ponatinib have demonstrated efficacy in CML and Ph+ ALL, their use in BCR-ABL1-associated AML has not been established. Please provide any in vitro data. (3) It has been suggested that combination therapy of TKIs with hypomethylating agents and BCL-2 inhibitors such as venetoclax may improve outcomes in this rare Ph+ AML subset [26,28–30]. →Please provide more details and add more information. (4) AML with BCR-ABL1 gene rearrangement is a high-risk disease that requires targeted treatment strategies due to its aggressive nature and poor response to standard AML therapies [27,32]. →Please provide more details and add more information.
